# Multiple Myeloma Insights from Single-Cell Analysis: Clonal Evolution, the Microenvironment, Therapy Evasion, and Clinical Implications

**DOI:** 10.3390/cancers17040653

**Published:** 2025-02-14

**Authors:** Sihong Li, Jiahui Liu, Madeline Peyton, Olivia Lazaro, Sean D. McCabe, Xiaoqing Huang, Yunlong Liu, Zanyu Shi, Zhiqi Zhang, Brian A. Walker, Travis S. Johnson

**Affiliations:** 1Indiana Bioscience Research Institute, Indianapolis, IN 46202, USA; 2Richard M. Fairbanks School of Public Health, Indiana University, Indianapolis, IN 46202, USA; 3School of Medicine, Indiana University, Indianapolis, IN 46202, USA; 4Regenstrief Institute, Indianapolis, IN 46202, USA; 5Melvin and Bren Simon Comprehensive Cancer Center, Indiana University, Indianapolis, IN 46202, USA; 6Center for Computational Biology and Bioinformatics, Indiana University, Indianapolis, IN 46202, USA

**Keywords:** myeloma, multiple myeloma, single-cell technologies, scRNA-seq, single cell, omics, clonal evolution, tumor microenvironment, personalized medicine

## Abstract

Multiple myeloma (MM) is a blood cancer of bone marrow that is difficult to diagnose and treat because it changes over time and has complex interactions with the body’s immune system. This review explores how single-cell technologies help scientists understand multiple myeloma more comprehensively by identifying hidden tumor clones, tracking how the disease evolves, and discerning why some treatments stop working. Researchers have found that certain genetic changes, like mutations or deletion of TP53, loss of chromosome 13, and gain of chromosome 1, make the cancer more resistant to treatment, and that the immune system often fails to fight off MM because there is a suppressive environment around the tumor. New treatments, like Chimeric Antigen Receptor T cell (CAR-T) cell therapy and Bispecific T cell engagers (BiTE), take advantage of the immune microenvironment and are showing promise in targeting myeloma cell directly. However, they are not always effective and can have adverse effects. By studying single-cell resolution data to identify new vulnerabilities in myeloma cells and pathways related to resistance that can be different within the cells in a single patient, new treatment regimens can be identified and tailored for individual patients.

## 1. Introduction

Multiple myeloma (MM) is a common blood cancer defined by the buildup of malignant plasma cells (myeloma cells), typically occurring in the bone marrow [1]. In this condition, myeloma cells produce abnormal proteins, e.g., immunoglobulins (Ig), and cause severe complications such as kidney damage, anemia, hypercalcemia, and bone weakening. Each year, over 30,000 individuals in the United States and over 150,000 people globally are diagnosed with MM [2]. MM typically involves chromosomal abnormalities and translocations [3,4]. About half of the patients have primary translocations involving one of five chromosomal partners (4, 6, 11, 14 or 20), specifically oncogenes juxtaposed to the immunoglobulin heavy chain (IgH) locus, and almost all cases of MM exhibit dysregulation of one of the three cyclin D genes [3,4]. Though some patients can have early onset disease, MM predominantly affects older adults, with an average onset age of 69 years [5]. As life expectancy continues to rise, it is critical to address MM treatment due to the resulting increase in prevalence of MM [6] by better understanding its molecular etiology.

The many unanswered questions and challenges in myeloma research involve improving dynamic disease risk assessment through MRD or other means, better defining high-risk disease, improving our understanding of the molecular basis of myeloma, improving guidance of treatment strategy, e.g., through new molecular markers, and by identifying novel targets, i.e., novel antigen or small molecule inhibitor targets [7,8]. Single-cell molecular profiling offers a detailed understanding of how MM impacts individual cells and can be used to address these many unanswered challenges at unprecedented resolution.

## 2. Single-Cell Analysis Technologies in MM

Single-cell profiling provides a more nuanced look at how MM affects cellular processes within myeloma cells and cells in the microenvironment. The understanding of MM development on a cellular level can also potentially provide insights into treatment through the identification of biomarkers that may answer why many patients become resistant to anti-myeloma drugs. With bulk RNA-seq and microarray, the averages for cellular populations are examined by aggregating a large volume of cells from a given sample. However, this can limit our knowledge on rarer cell types, such as cancer stem cells or immune cell subsets, which may have more insight into many of the questions of MM biology [9]. Similarly, the generalization of results across these large bulk samples can have a confounding effect and makes it difficult to determine how MM clones can evolve over time, and how these clones are able to resist treatment. The use of scRNA-seq in this field has been developed over time, firstly with mRNA transcriptomics in 2009, which allowed for the large-scale detection of mixed-cell samples, followed by the introduction of Smart-seq in 2012 (Figure 1), which enhanced the read coverage of scRNA-seq and enabled the analysis of alternative splicing and single nucleotide variant (SNV) detection, which was later refined in Smart-seq2 [10,11]. Notably, Fluidigm C1 microfluidic chips in combination with multiplexed qPCR and Smart-seq2 have already been used in MM research [12,13]. MARS-seq, an approach that barcodes cells using a multiwell plate with individual cells in each well, was also commonly used in MM research [14,15]. These techniques now have largely been superseded by the much higher throughput droplet barcoding approaches such as the 10x Genomics Chromium platform (Figure 1, Table 1) [1,16,17,18,19,20,21,22,23,24,25,26,27,28,29,30,31,32,33,34,35,36,37,38].

The development of single-cell proteomics protocols for MM research has largely relied on either a basis of mass spectrometry or targeted antibodies. The most popular of these techniques used in MM are generally antibody-based including 10x Genomics multiome RNA + Protein [33], CITE-seq [31], and Mission Bio Tapestry DNA + Protein [40]. However, the combination of the mass spectrometry approach and flow cytometry has created mass CyTOF, which allows for the study of specific proteins in single cells through isotope-labeled antibody conjugation with specific molecules in or on the cell surface (Figure 1). Protein expression data can be used by itself or as a validation method for differential expression analysis between cell subpopulations defined from scRNA-seq to evaluate the consistency between differentially expressed RNAs and proteins [30].

Increasingly, MM research has begun to use newly developed single-cell genomics approaches such as Mission Bio Tapestry, which uses novel probe sets to analyze SNVs, SNVs and copy number variants (CNVs), and DNA + Protein [40]. Alternatively, some groups have used 10x Genomics protocols to perform single-cell genomics profiling [1] or single-cell multiplexed qPCR to identify SNV and CNVs [13]. All of these approaches have provided ample resources for the study of genetics, genomics, epigenetics, and proteomics at the single-cell level in myeloma.

## 3. Subclonality and Tumor Evolution

### 3.1. Genomic Instability Drives Subclonal Diversity in MM

The clonal architecture of MM is highly dynamic, marked by distinct subclones that evolve in response to selective pressures, including therapy and microenvironmental interactions. This subclonal architecture is evident even in precursor diseases, such as monoclonal gammopathy of undetermined significance (MGUS) and smoldering myeloma and persists throughout disease progression to relapsed/refractory MM (RRMM) [32,42,43]. Subclones often exhibit unique genetic alterations, such as mutations in *NRAS*, *KRAS*, and *TP53*, which occur as secondary events that contribute to disease progression, treatment resistance, and relapse [43,44,45].

Genomic instability is a hallmark of MM and plays a crucial role in driving subclonal diversity and disease progression. It encompasses both genome-wide changes, such as aneuploidy and chromosomal translocations, and micro-level changes, including SNVs and DNA repair defects [32,43,46]. Notably, alterations in the *TP53* gene, including deletions and SNVs, are among the most well-studied genomic abnormalities in MM, frequently associated with poor prognosis, increased clonal fitness, and resistance to therapy [29,45,47].

The impact of genomic instability on MM progression is evident in its ability to drive the emergence of drug-resistant subclones. As myeloma cells acquire genomic alterations, they develop mechanisms to evade treatment, including alterations in DNA damage repair pathways, epigenetic reprogramming, and force changes in the bone marrow microenvironment [32,44]. High-risk genomic features, such as biallelic *TP53* alterations, gain(1q), and translocations involving the immunoglobulin heavy chain locus, are known to drive treatment resistance in some studies [32,42,45]. Moreover, the presence of multiple, genetically distinct subclones complicates treatment, as different subclones may harbor resistance mechanisms against various therapies, such as Bortezomib [48], necessitating a more personalized approach [19,44] (Figure 2).

Recent single-cell multi-omics analyses have revealed that multidrug-resistant subclones often co-exist within the same patient, each with unique resistance mechanisms, such as altered drug transport, metabolic reprogramming, and immune evasion [32,43]. This parallel development of resistance highlights the need for combination therapies that can simultaneously target multiple subclones and their resistance pathways [29,46]. Indeed, most patients are now treated with combination therapies such as lenalidomide, bortezomib, and dexamethasone (RVD) as induction therapy regardless of transplant eligibility [49], carfilzomib, melphalan, and prednisone (KMP) for transplant ineligible NDMM patients [50], carfilzomib, cyclophosphamide, lenalidomide, and dexamethasone (KCRD) for NDMM patients [51], bortezomib, dexamethasone, cisplatin, doxorubicin, cyclophosphamide, and etoposide (VD-PACE) for triple-class refractory RRMM [52], daratumamab (i.e., a CD38 monoclonal antibody) combination therapies such as D-RVD [53], D-KRD [54], and numerous others depending on patient characteristics, especially in RRMM [55]. Quad therapies are now standard treatments for MM patients.

### 3.2. Tumor Evolution in Myeloma

Mutagenesis in MM is the result of natural processes of aging, defects in DNA repair mechanisms, and the mutagenic activity of enzymes like activation-induced deaminase (AID) and APOBEC. The specific mutagenesis processes differ depending on the primary genomic lesions, with HRD MM showing signatures associated with aging, and MM with translocations showing signatures related to APOBEC activity and DNA repair defects [56,57]. The main selective pressures for MM evolution include competition for resources and treatment while the mechanisms driving tumor evolution in MM include chromosomal translocations, CNVs, SNVs, epigenetic modifications, interclonal interactions, and microenvironmental influences [56,57].

The earliest events in MM pathogenesis are usually chromosomal translocations involving the immunoglobulin heavy chain (IgH) locus on chromosome 14q32 [58]. These in turn cause expansion of a single Ig clone resulting in a very heavily overrepresented antibody type, usually IgG or IgA (Figure 3). Notable translocations include t(4;14), t(11;14), t(6;14), t(14;16), and t(14;20), which are implicated in distinct molecular mechanisms driving MM progression [4,59]. These translocations lead to the deregulation of oncogenes such as *FGFR3*, *NSD2*, *CCND1*, *CCND3*, *MAF*, and *MAFB*, respectively, resulting in aberrant cell cycle control, increased proliferation, resistance to apoptosis, and ultimately contributing to distinct clinical behaviors and prognostic outcomes [4,58,60,61,62,63,64,65,66]. In addition, dysregulation of epigenetics, including DNA methylation and histone modifications, has emerged as a crucial regulator of MM pathogenesis [67].

Over 80 driver genes are repeatedly mutated in MM patients, and a significant percentage of these SNVs are frequently subclonal, indicating they develop later in the course of tumor evolution than the primary initiating event. The acquisition of SNVs in driver genes is not a random process and depends on the primary genomic event initiating oncogenesis in MM and leading to the expansion of more aggressive or drug-resistant clones under selective pressures, such as therapy or microenvironmental factors [68,69,70]. This results in a heterogeneous population of subclones with distinct genetic profiles, each having different growth advantages and responses to treatment. Subclonal heterogeneity is a key factor in treatment resistance and disease relapse, as different subclones may respond differently to therapy, leading to an evolving and adaptive tumor landscape [71,72].

Clonal evolution, where distinct genetic subclones emerge over time, contributes to disease heterogeneity and can lead to variable treatment responses [73]. Clonal evolution can lead to a “phenotypic shift”, which can limit the ability to detect measurable residual disease (MRD) using PCR-based methods but has less of an impact on higher resolution flow cytometry and next-generation sequencing MRD techniques [74]. As the disease progresses, specific biomarkers such as chromosomal abnormalities and elevated serum B-Cell Maturation Antigen (sBCMA) levels become instrumental in predicting patient outcomes [75,76]. For example, higher sBCMA levels correlate with severe disease and can signal shorter progression-free and overall survival [77,78]. Cytogenetic aberrations have a greater prognostic impact in MM than mutations in specific genes. The evolution of cytogenetic abnormalities, particularly the acquisition of high-risk aberrations like del(17p), adversely affects patient prognosis [77]. Though the mechanism is still not fully understood, gain/amp(1q) is one of the highest risk cytogenetic aberrations and may operate through upregulation of *CKS1B*, *MCL1*, *BCL9*, and/or *PBX1* [26,79,80]. Other high risk cytogenetic events include del(1p) resulting in loss of *CDKN2C*, del(13q) resulting in loss of *RB1*, and multiple others [81]. Spatial heterogeneity of MM in different bone marrow lesions can lead to the missed detection of prognostically significant cytogenetic aberrations in a significant percentage of cases [82]. While single 1q abnormalities have been widely studied in multiple myeloma, recent analyses suggest that patients harboring double- or triple-hit cytogenetic abnormalities, such as concurrent 1q amplification, del(17p), and t(4;14), exhibit significantly poorer prognoses. These high-risk patients tend to exhibit elevated genomic instability, making their disease more aggressive. As a result, they often respond poorly to standard therapies, and thus require alternative treatment approaches and more vigilant clinical monitoring [83].

In MM, the development of targeted therapies and personalized medicine should be guided by the tumor’s genetic heterogeneity and evolutionary dynamics, which vary due to both intratumor and intertumor heterogeneity (Figure 4A). These complexities highlight the need for treatment strategies that can adapt to MM’s clonal evolution, which is dependent on various factors including patient- and tumor-related features [84]. Personalized treatment strategies increasingly incorporate insights into genetic diversity, requiring therapeutic approaches that address both the inter- and intra-patient variability driven by tumor evolution [85]. As a result, dynamic therapy models such as sequential and adaptive therapies have been developed to modulate treatment in response to changes in the tumor’s genetic profile, targeting the tumor’s evolution to potentially achieve sustained disease control. These models consider the shifting genomic landscape, offering a framework for long-term management of MM by adjusting treatments based on real-time disease progression [86].

Up until now, there has only been one genomic alteration, t(11;14), that is regularly used for precision medicine, i.e., to define patient cohorts that are likely responsive to venetoclax [87]. There have been recent proposals to use liquid biopsy methods in precision medicine. These include the analysis of circulating tumor cells (CTCs) and cell-free DNA (cfDNA) to provide a novel tool for tracking MM’s clonal evolution in a noninvasive manner [88,89] (Figure 4B). Unlike single-site bone marrow biopsies, this method could offer a less spatially biased view of the subclonal architecture of the disease, allowing for the detection of new mutations or clonal expansions that might otherwise go unnoticed [90,91]. On the other hand, these assays could also miss myeloma cell clones that are predominantly localized in the bone marrow. Though there are clear limitations to liquid biopsies, their use could help identify circulating clones irrespective of spatial heterogeneity within the bone marrow [92]. Whether bone marrow biopsies, bone marrow aspirates, or liquid biopsies are used, ongoing clonal monitoring should be used to adapt treatment regimens to the shifting genetic profile of MM in the future [93,94] (Figure 4C).

## 4. Interplay Between Subclones and the Microenvironment

The interplay between subclones in MM is characterized by both competitive and cooperative interactions that drive disease evolution [20,29,32,34,44,95,96,97,98,99,100,101,102,103]. Subclonal competition, influenced by genetics and environmental pressures, leads to the expansion of dominant clones and the emergence of treatment-resistant populations [102]. However, subclones may also exhibit cooperative behavior, where interactions between them and with the tumor microenvironment facilitate mutual survival and adaptation [20,34,70].

### 4.1. Subclone Competition

Subclonal competition is driven by the tumor’s inherent genetic instability, which results in diverse subclonal populations with different survival advantages [79]. Treatment provides selective pressure and leads to complex clonal dynamics, where the relative abundance of subclones shifts over time in response to therapeutic interventions, resource competition, and proliferative advantages [20,101]. The competitive interactions among subclones in MM can be explained by several theoretical models of evolutionary dynamics. Branching evolution, for instance, suggests that multiple subclones arise independently and co-exist, each competing for resources within the tumor microenvironment [95,97]. A whole exome sequencing study [97] highlights the diversity within the tumor and explains why certain subclones may become dominant in response to specific pressures, such as the availability of resources or immune evasion mechanisms [20,29,34]. In contrast, linear evolution suggests a more straightforward pathway where one dominant clone sequentially gives rise to another, reflecting a continuous adaptation of the dominant subclone to the microenvironmental changes [102].

Subclonal competition also results in the emergence of treatment-resistant clones, particularly in advanced disease stages such as relapsed/refractory multiple myeloma (RRMM). For example, single-cell RNA sequencing (scRNA-seq) has revealed that patients with RRMM often develop resistant clones characterized by specific genetic mutations, such as gain(1q), which is linked to a unique microenvironment, treatment resistance, and poor prognosis. Through joint scRNA-seq study of myeloma cells and the microenvironment, gain(1q) clones were associated with increased numbers of specific subtypes of tumor associated macrophages that express *C1QA* and *CD206*, CD56^dim^ natural killer (NK) cells, and a subset of inflammatory dendritic cells (DC) expressing *S100A8*, *S100A9*, and *CD14*, *CLEC7A*, and *VSIR* [34]. The expansion of these resistant clones is associated with the upregulation of genes related to survival, cell proliferation, and immune evasion, allowing them to thrive in conditions where other clones cannot [34]. Another study using single-cell transcriptomics and epigenetics characterized the subclonal-level resistance mechanisms [32]. In this study, resistance mechanisms could cooccur across subclones such as an increase in Heat Shock Proteins (HSPs), i.e., HSP90AA1, after treatment with carfilzomib or an increase in NF-κB pathway after treatment with MEK/BRAF inhibitors [32]. Resistant subclones also had different interactions with the microenvironment [32]. The complexity of subclonal competition underscores the need for personalized treatment strategies that can adapt to the evolving clonal landscape throughout a patient’s disease course, as the dominance and prevalence of subclones are not static but are instead constantly shifting in response to therapeutic and environmental pressures [95,103].

### 4.2. Subclone Cooperation

Interestingly, subclones in MM do not merely compete; they may also exhibit cooperative behaviors that accelerate disease progression. This phenomenon of clonal cooperation is observed when subclones interact with each other and with the tumor microenvironment to foster mutual survival or the formation of new clones [99]. Studies have shown that plasma cell extrinsic factors through subclones may contribute to the acquisition of invasive phenotypes and facilitate the progression of the disease from precursor stages, such as monoclonal gammopathy of undetermined significance (MGUS) and smoldering multiple myeloma (SMM), to symptomatic multiple myeloma [103]. Additionally, studies have shown that clonal cooperation plays a significant role in tumor formation and the acquisition of invasive phenotypes, indicating that subclones can indeed help each other evolve in certain contexts [101]. However, it is not clear how the subclones work together over time [96] through these complicated dynamics, where cooperation and competition coexist to drive tumor progression. Time series single-cell studies will be essential to understand which subclones persist together in a cooperative state.

### 4.3. Inter-Subclone Communication

During MM progression, subclones may communicate, signal, or stimulate one another. The mechanisms for this are still not well defined or understood. One example of possible inter-sublcone communication is in endogenous IL6 production by some myeloma cells where the IL6 production from the myeloma cells can have a self-stimulatory effect causing increased growth [104,105]. Alternatively, CD44 can promote cellular adhesion amongst myeloma cells and with the extracellular matrix (ECM), resulting in resistance to Lenalidomide [106]. In relapsed/refractory multiple myeloma, subclones that have acquired resistance to treatment often engage in interactions that further alter the tumor microenvironment, such as the production of inflammatory cytokines or adhesion molecules, which can create a pro-inflammatory environment conducive to tumor growth [34]. However, as certain subclones achieve dominance, there may also be a reduction in inter-subclone communication. This reduction in inter-subclone communication may lead to a more homogeneous tumor population, at least temporarily, until new selective pressures prompt further diversification and communication among emerging resistant subclones [102]. Nevertheless, the underexplored dynamic nature of inter-subclone communication highlights the current challenge of developing durable therapeutic strategies [20,32,98] and underscores the adaptability of subclonal populations in response to their changing microenvironment and treatment landscape.

### 4.4. Subclone and Microenvironment Interactions

The myeloma microenvironment is a complex network of immune and stromal cells that can support tumor growth and can contribute to therapy resistance (Figure 5A). Single-cell studies have increasingly shown that the immune microenvironment is directly affected by genetic changes in the myeloma cells and co-evolves along with them [30]. With the generation of large single-cell datasets of the myeloma microenvironment, the complex relationships and dynamics of the cells are increasingly well understood. This includes how myeloma cells may affect their immune microenvironment. Myeloma cells can induce dysfunction in T cells by expressing immune checkpoint molecules like PD-L1, leading to T cell exhaustion and reduced anti-tumor activity. This is accompanied by a decrease in CD4+ T cells, further weakening the immune response against myeloma cells [107,108,109,110]. Furthermore, single-cell studies have shown that T-cells from MM patients tend to shift from a cytotoxic toward an effector phenotype and that these changes can be variable between patients representing large amounts of heterogeneity both within and between patients [38]. Regulatory B cells (Bregs) in the myeloma microenvironment facilitate immune tolerance by secreting transforming growth factor-β (TGF-β), interleukin-10 (IL-10), and interleukin-35 (IL-35), suppressing anti-tumor immune responses [20,21,22]. Macrophages, often polarized to an M2 phenotype, i.e., TAMs, in MM, secrete growth factors and cytokines that promote tumor growth and suppress immune responses [34]. DCs are crucial for antigen presentation and T-cell activation, yet their function is impaired in the myeloma microenvironment, further suppressing the immune response [35]. These DCs can be activated using an anti-CD40 agonist to reduce myeloma cell load and other symptoms [35]. Single-cell studies of the immune microenvironment have shown that all these cell types change in proportion between healthy donors, NDMM, and RRMM [20]. Notably, MM patients had increased numbers of NK cells, CD16+ monocytes, and CD8+ memory/effector cells compared to healthy donors [20]. After treatment, the CD14+ monocytes also increased significantly in proportion and began expressing chemokines including *CCL3*, *CCL4*, and *CXCL2* [20]. In contrast, CD16+ monocyte pretreatment mainly expressed IFN alpha response pathway genes including *IFI6*, *IFL44L*, and *IFITM* [20]. Taken together, there are clear immune compositional and phenotypic shifts in MM and during treatment.

The stromal cells in the microenvironment, such as fibroblasts, osteoclasts, and endothelial cells (ECs), provide structural support and contribute to disease progression. Fibroblasts often differentiate into cancer-associated fibroblasts (CAFs), which secrete factors that promote tumor growth and survival [111]. Single-cell studies have identified subtypes of CAFs that express platelet markers including *MPL* and *ITGA2B* that may mediate growth [41]. Myeloma cells stimulate osteoclast activity, leading to bone resorption and release of growth factors that further promote tumor growth [109,112]. From single-cell studies of the primary site of aberrant osteoclast activity, subclones were found to localize to focal lesions and have altered expression such as downregulation of *CXCR4* [25]. ECs respond to pro-angiogenic factors, such as VEGF produced by myeloma cells and other microenvironmental cells, by forming new blood vessels. This increased vascularization delivers essential nutrients and oxygen, fostering tumor growth and facilitating metastasis [111,113]. Subtyping analysis and comparison of ECs between MM stages using scRNA-seq revealed a pre-vascular to angiogenic shift in ECs from MGUS to MM, an increase in IFN signaling in subsets of ECs in MM that may represent a unique MM subtype, and transcriptional changes associated with angiogenesis, migration, lipid metabolism, and increased expression of *SOX18* during MM progression [17]. ECs and the resulting angiogenesis signaling and cytokine release can be closely linked in the MM microenvironment.

To further elaborate on the role of the tumor microenvironment in MM resistance, we highlight key cell types influencing disease progression. Mesenchymal stromal cells (MSCs) provide structural support and contribute to a pro-tumorigenic niche by secreting cytokines and extracellular matrix components that improve myeloma cell survival and therapy resistance [114]. Dysfunctional T-cell immunity, particularly T-cell exhaustion and impaired cytotoxicity, allows for immune evasion and disease progression in MM [115]. M2 macrophages contribute to an immunosuppressive tumor microenvironment by secreting anti-inflammatory cytokines, reducing antitumor immune responses [116]. Vascularization, i.e., enhanced angiogenesis, driven by VEGF and other pro-angiogenic factors, promotes nutrient delivery to myeloma cells, facilitating their proliferation and drug resistance [117]. These factors actively influence the clonal evolution of MM, thus contribute to the intricate and dynamic interactions within its microenvironment.

Cytokine signaling pathways and immune evasion mechanisms are central to the interactions between myeloma cells and their microenvironment. Key cytokines such as interleukin-6 (IL-6), TGF-β, and IL-10, produced by immune and stromal cells within the tumor microenvironment, play a critical role in promoting tumor growth and creating an immunosuppressive niche. IL-6 is particularly important in enhancing myeloma cell proliferation and survival, as well as supporting drug resistance [118,119]. TGF-β suppresses T-cell function, enables immune evasion, and facilitates tumor metastasis [110]. Subtyping of T-cells via scRNA-seq has further demonstrated a significant increase in *GZMK* and *TIGIT* expressing exhausted CD8 T-cells in MM patients who rapidly progressed [31], whereas patients who did not progress had increased numbers of immature B cell subsets expressing *IGLL1*, *SOX4*, and *DNTT* [31]. Meanwhile, IL-10, a potent immunosuppressive cytokine, supports tumor progression by enhancing myeloma cell survival, dampening anti-tumor immune responses, and promoting resistance to therapy [120].

### 4.5. Immunotherapies

Considering these challenges, innovative therapeutic strategies have been developed to target the myeloma microenvironment, aiming to disrupt these supportive interactions or facilitate immune response. Chimeric antigen receptor T (CAR-T) cells (Figure 5B), engineered to target specific antigens on MM cells, such as B-cell maturation antigen (BCMA), CD38, or G-protein-coupled receptor family C group 5 member D (GPRC5D), have demonstrated significant efficacy in patients with relapsed or refractory MM [121,122]. Upon antigen recognition, CAR-T cells release cytotoxic molecules, including granzymes and perforins, leading to myeloma cell death and effective tumor elimination. However, the immunosuppressive nature of the myeloma microenvironment presents a substantial challenge to CAR-T cell durability and efficacy. T cells within this environment are often exposed to sustained antigen stimulation and inhibitory cytokines, leading to T-cell exhaustion, a state in which CAR-T cells lose functionality and fail to persist long-term in the patient. A single-cell analysis of BCMA CAR-T cell therapy identified increased numbers of CD8 T-cells and NK cells and decreased CD14+ monocytes in BCMA CAR-T cell therapy responders compared to non-responders with consistently higher expression of PIM kinases in the monocytes, DCs, and NKs from non-responders [33]. Furthermore, this analysis also revealed CD39 as a potential target either before or after BCMA CAR-T cell therapy [33]. Researchers are exploring alternatives to BCMA CAR-T cell therapy via alternative antigens (GPRC5D or CD38) [123] and the development of dual-targeted CAR-T cells that recognize multiple antigens, such as BCMA and CD19 [124]. In addition, a recent study demonstrated that armored BCMA CAR-T cells with dominant-negative TGF-β receptors resist TGF-β suppression, maintaining their cytotoxic function within the immunosuppressive myeloma microenvironment [125]. These strategies will allow clinicians to provide more options so that they can more effectively select CAR-T cell therapies as the tumor adapts.

Bispecific antibodies have emerged as another potent tool for targeting myeloma cells using the microenvironment (Figure 5C), offering an “off-the-shelf” option that bypasses the need for cell modification [126]. These antibodies effectively bridge T cells and MM cells by simultaneously binding CD3 on T cells and specific antigens on MM cells, such as BCMA, CD38, Fc receptor-homolog 5 (FcRH5), or GPRC5D. This interaction activates T cells, prompting the release of granzymes and perforins that induce MM cell death. By leveraging the immune system, this therapeutic strategy enhances T cell-mediated cytotoxicity and disrupts the immunosuppressive tumor microenvironment. Teclistamab is a notable bispecific antibody targeting BCMA and CD3 that has demonstrated high response rates in clinical trials, particularly in patients with heavily pre-treated and refractory disease [127]. Unlike CAR-T cell therapies, which require patient-specific genetic modification, bispecific antibodies are more readily available and can be administered directly, providing an immediate and effective response. Studies have shown that bispecific antibodies not only enhance T-cell recruitment to the tumor site but also activate T cells directly within the bone marrow microenvironment, circumventing some of the suppressive elements that CAR-T cells encounter [126,128]. In a single-cell multiomic study of BMCA BiTE-treated patients, T cell receptor (TCR) diversity decreased, there was an increase in a subset of exhausted T cell, and *BCMA* expression decreased in poor responders after treatment [21]. These advanced therapies aim to reshape the microenvironment, reinvigorating the immune response and overcoming the protective role that stromal and immune cells provide to myeloma cells. By targeting the interactions between the tumor and its microenvironment, these approaches offer the potential for more durable and effective treatments for MM.

## 5. Clinical Utility of Single-Cell Technologies in the Myeloma

Research over the past decade has shown that providing effective clinical applications and precision medicine to patients with MM is significantly challenging because of the heterogeneity between patients and within tumors [15], as well as the dynamic clonal evolution characteristic of this disease [30]. Single-cell technologies like scDNA-seq, scRNA-seq, and emerging methods such as scATAC-seq and single-cell proteomics offer a superior approach for further studying the complexity and clinical treatment of MGUS, SMM, and MM (Figure 6A). In fact, the single-cell resolution technique flow cytometry is already used to measure MRD [129]. The clinical translation of single-cell profiling is facilitated by its capacity to analyze the dynamics and heterogeneity of both tumor and immune cells, delve into the interactions within the tumor microenvironment, and detect treatment responses and drug resistance amid clonal evolution. The growing body of research and expanding datasets underscore the substantial potential of single-cell technologies to improve diagnosis, prognostication, and monitoring of treatment response and residual disease in MM patients.

Single-cell and multi-omics sequencing have greatly enhanced the ability to study spatial genetic heterogeneity within a single location and between distant tumor locations. Datasets of scRNA-seq have been utilized to investigate the dynamics of tumor heterogeneity throughout the progression of MM [12,15,16,26]. For example, scRNA-seq was used to examine the heterogeneity among 40 individuals across the progression spectrum of MM [15]. In the study, asymptomatic individuals with early-stage disease and those with minimal residual disease post-treatment detected rare tumor plasma cells exhibiting molecular characteristics akin to those of active myeloma [15], suggesting potential implications for personalized therapies. Unique B cell lineages can be annotated from scRNA-seq with seven continuous B lymphocyte lineages annotated in one study. MM affects the development of B cells and the heterogeneity of plasma cells that could be targeted via WNK1 [36].

Single-cell sequencing is also a potent tool to evaluate clonality (Figure 6B) and compositional and phenotypic shifts in the microenvironment (Figure 6C), enabling high-resolution mapping of dysregulation occurring between disease stages. Changes in immune cell composition between SMM, NDMM, and MM are associated with disease progression and potential mechanisms of immune evasion [38]. Additionally, myeloma development and progression in the microenvironment are influenced by both clonal competition and cooperation, occurring within the context of dynamic changes in both the cellular and noncellular components of the tumor microenvironment (Figure 6D). By integrating scRNA-seq with various genomic platforms, the clonal evolution and the impact of the tumor microenvironment could be assessed using longitudinal samples from patients [19,21]. These same samples could be used to track plasma cell subpopulations at different stages of disease and to discover patient-specific plasma cell profiles and immune cell expression changes [30]. High-throughput single-cell DNA sequencing of circulating CD34+ cells from MM patients revealed clonal hematopoiesis of indeterminate potential (CHIP)-associated mutations in 50% of cases, with the most common mutations in *TET2*, *EZH2*, *KIT*, *DNMT3A*, and *ASXL1* [39]. These longitudinal samples showed a selection of high-fitness mutant clones over time, indicating a trend toward suboptimal therapy responses in CHIP-positive patients [39].

MM therapeutics apply considerable selective pressure that influences the clonal evolution of myeloma cells. Single-cell profiling contributes to identifying low-frequency tumor clones in asymptomatic diseases, detecting drug-resistant clones during treatment and identifying signatures of treatment response and resistance. Using single-cell profiling, the pattern of clonal evolution has been shown to mirror the response to antimyeloma therapy. While MM treatment eradicates sensitive subclones, it typically promotes the selection of drug-resistant subclones that were either initially present in the tumor mass or emerged during therapy [97]. Treatment resistance occurs either from the selection of preexisting subclones, e.g., cells constituting MRD, or acquisition of new subclonal genetic changes [130]. Datasets of scRNA-seq have been applied to study drug resistance mechanisms in highly resistant MM patients. The prospective, multicenter, single-arm clinical trial (NCT04065789) combined longitudinal scRNA-seq and clinical data to study the molecular dynamics of MM resistance mechanisms. This study defined a roadmap for integrating scRNA-seq in clinical trials, identified a signature of highly resistant MM patients, and discovered peptidylprolyl isomerase A (PPIA) as a potent therapeutic target for these tumors [14]. Furthermore, scRNA-seq has been used for MM drug discovery. The computational pipeline “secDrugs” employs pharmacogenomics data to optimize and regularize a greedy algorithm for predicting new drugs targeting drug-resistant myeloma. The algorithm utilized scRNA-seq as a screening method to identify top combination drug candidates, focusing on the enrichment of target genes [27].

Single-cell profiling has proven effective as a transformative technology in research, but numerous challenges must be addressed to operationalize this technology in clinical settings (Figure 6E). First, the costs associated with single-cell assays are prohibitively high for widespread routine clinical use, including sequencing, specialized equipment, and reagents [131]. Second, like any new technology, single-cell technologies’ reliability, reproducibility, and clinical relevance need rigorous validation. Third, the sensitivity and prevalence of genetic events identified through single-cell technologies must first be determined before they can be implemented in clinical practice, otherwise the treatment strategies will not be effective. Fourth, the lack of standardized bioinformatic pipelines from sample preparation to data analysis leads to variability and makes it difficult to compare findings across studies and to develop standardized tests for clinical use [132]. Furthermore, the data generated from single-cell technologies are complex and require advanced computational tools and analysis expertise. The complexity of data analysis can be a significant barrier in clinical settings, which may not have the necessary computational infrastructure or bioinformatics support. Last, there would be many ethical and regulatory challenges [133], particularly concerning privacy and the management of the large amounts of genetic data generated, and a lengthy process for regulatory evaluation and approvement.

## 6. Discussion

This review highlights how single-cell technologies have helped to unravel the complexities associated with MM (Table 1), particularly its clonal evolution, microenvironmental interactions, and clinical applications. The dynamic nature of clonal evolution and the substantial genetic and phenotypic heterogeneity present in MM underscore the challenges faced in diagnosing and treating the disease. Understanding the evolution of subclones, the selective pressures from therapeutic interventions, and the tumor microenvironment is crucial in determining effective personalized treatment approaches. Notably, genomic instability and mutations in key oncogenes like *TP53*, *KRAS*, and *NRAS* significantly contribute to treatment resistance, emphasizing the need for novel targeted therapies.

The myeloma microenvironment plays a pivotal role in disease progression and resistance by fostering immune evasion and providing structural and biochemical support for tumor growth. The involvement of immune cells, including exhausted T cells, TAMs, and regulatory B cells, creates an immunosuppressive environment that favors tumor survival. Targeting the myeloma microenvironment through innovative therapeutic approaches like CAR-T cell therapy and BiTEs offers promising avenues for overcoming these challenges and enhancing treatment efficacy.

The incorporation of single-cell technologies into the study of MM, such as scRNA-seq and single-cell genomics, has provided new insights into the complexity of MM at the highest levels of granularity, enabling the dissection of intratumor heterogeneity and the identification of rare but clinically relevant cell populations. It would also be valuable to investigate whether aberrant clones exist prior to treatment or emerge as a result of therapy. The occurrence of acute myeloid leukemia in some myeloma patients, and vice versa, suggests the presence of a mutation in an early stem cell. However, relapsed or refractory patients have increased incidence of high-risk mutations and cytogenetics alluding to selection pressure from treatment. This remains an unresolved issue in the field of myeloma, but single-cell technology for patients at multiple time points during progression and treatment could provide insights to help address it. These technologies allow for a deeper understanding of clonal dynamics, interactions within the tumor microenvironment, and mechanisms of drug resistance, facilitating more precise identification of therapeutic targets. Despite the transformative potential of these technologies, the translation of these single-cell approaches into clinical practice is hindered by several challenges, including high costs, data complexity, the need for advanced bioinformatics, and ethical considerations.

To advance the field, future research should focus on addressing these barriers to the clinical implementation of single-cell technologies. Future prospective studies and clinical trials, such as NCT04065789, will play a crucial role in fully uncovering the impact of single-cell technologies in precision oncology. By integrating high-resolution single-cell data with comprehensive epidemiological, clinical, and imaging information, researchers can gain a more in-depth understanding of disease heterogeneity, the evolution of resistance mechanisms, as well as potential therapeutic targets. Developing cost-effective methods, standardizing bioinformatic pipelines, and establishing ethical frameworks for data management are crucial steps. Additionally, combining multi-omics approaches with single-cell profiling could further enhance our understanding of MM and provide a comprehensive view of disease progression, which is essential for developing truly personalized treatment strategies. A collaborative effort between researchers, clinicians, and bioinformaticians will be necessary to overcome these obstacles and fully harness the potential of single-cell technologies to improve outcomes for MM patients.

## 7. Conclusions

Single-cell technologies have helped to revolutionize our understanding of MM, especially how the disease changes over time and how it interacts with the tumor microenvironment. These new methods show that myeloma cells have evolving mechanisms to resist therapy and complex interactions with the tumor microenvironment leading to immune evasion that are driven by dynamic genetic changes. To date, MM treatment is not adequately personalized to patients based on high-risk genetics despite increasingly more studies showing links between these genetics and clinical outcomes. Even though single-cell technology is very useful, it is still not easy to use in real medical practice because of the costs, complex analysis, and privacy issues. In the future, scientists should work on making these technologies more accessible to practitioners to facilitate clinical adoption. Understanding myeloma one cell at a time has the potential to improve myeloma research and even clinical practice.

## Figures and Tables

**Figure 1 cancers-17-00653-f001:**
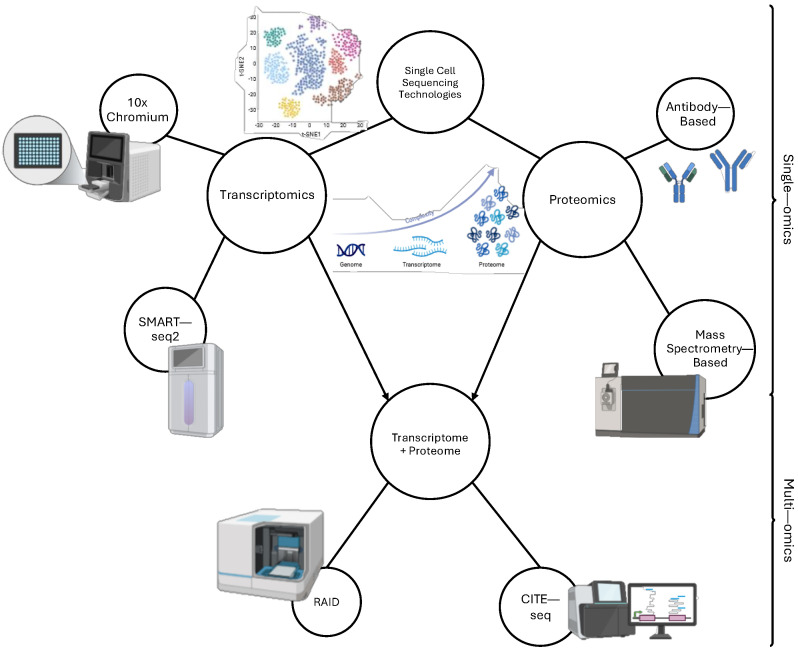
Different types of single−cell sequencing technologies that have been used for multiple myeloma research. Developments in single-omics such as transcriptomics (ex: 10x Chromium and SMART−seq2) and proteomics (ex: antibody-based vs. mass-spectrometry based) have led to integrative analyses in multi-omics, which combine multiple categories of single-cell technologies in methods such as CITE−seq and RAID.

**Figure 2 cancers-17-00653-f002:**
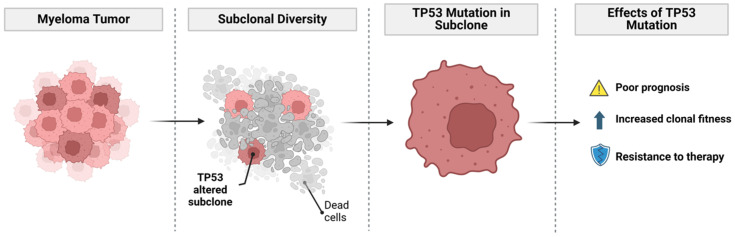
Representation of subclonal diversity and the role of TP53 mutations in multiple myeloma progression and treatment resistance. Created in BioRender. Peyton, M. (2025) https://BioRender.com/d94o089 (accessed on 11 February 2025).

**Figure 3 cancers-17-00653-f003:**
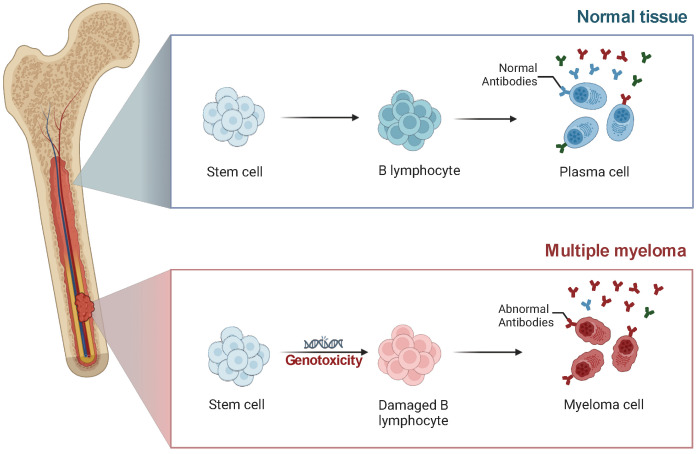
Stem cell differentiation and pathogenesis in multiple myeloma. Normal tissue: Stem cells differentiate into B lymphocytes, which further mature into plasma cells capable of producing normal antibodies, ensuring a functional immune response. Multiple myeloma: Genotoxic stress or other factors lead to damage in B lymphocytes. This damage disrupts normal differentiation, resulting in the formation of malignant myeloma cells that produce abnormal antibodies. These cells propagate within the bone marrow, contributing to the progression of MM. Created in BioRender. Liu, J. (2025) https://BioRender.com/w56y431 (accessed on 27 January 2025).

**Figure 4 cancers-17-00653-f004:**
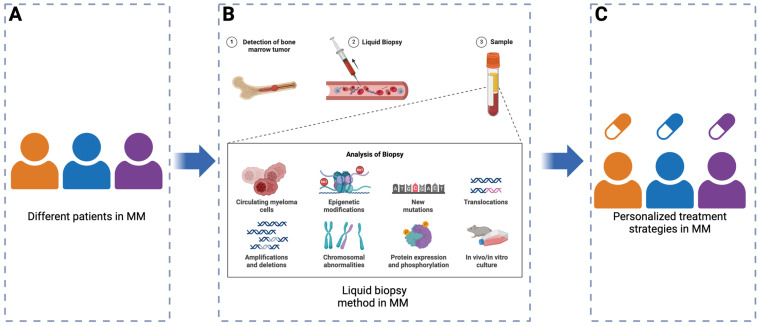
Liquid biopsy in multiple myeloma for personalized treatment strategies. (**A**) Different patients with MM represent the variability in disease presentation and progression. (**B**) Liquid biopsy enables non-invasive sampling and analysis of blood for circulating myeloma cells, genetic and epigenetic alterations, such as amplifications, deletions, chromosomal abnormalities, mutations, and translocations. It includes studies on protein expression and phosphorylation as well as in vitro culture for functional assessments. (**C**) Insights from liquid biopsy guide the development of personalized treatment strategies tailored to the molecular and cellular characteristics of each patient’s disease, improving therapeutic outcomes. Created in BioRender. Liu, J. (2025) https://BioRender.com/q52m439 (accessed on 27 January 2025).

**Figure 5 cancers-17-00653-f005:**
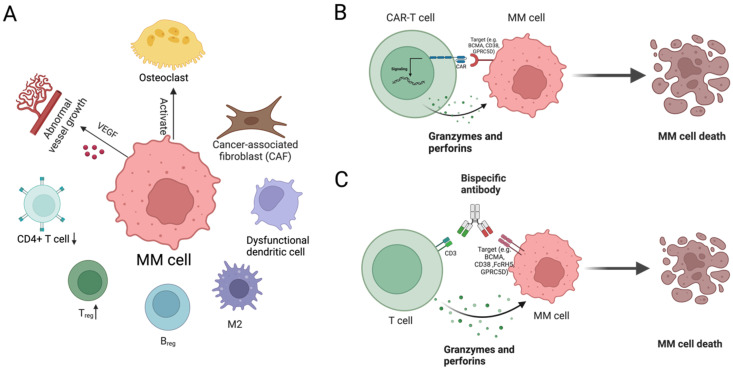
MM microenvironment and immunotherapeutic strategies. (**A**). Immune and stromal cells in MM microenvironment: The MM microenvironment includes T cell exhaustion (accompanied by decreased CD4+ T cells), increased regulatory T cells (Tregs), immunosuppressive regulatory B cells (Bregs), M2-polarized macrophages, and dysfunctional dendritic cells. Cancer-associated fibroblasts (CAFs) support tumor growth, myeloma cells stimulate osteoclast activity, and VEGF secreted by myeloma cells (and other microenvironmental cells) drives abnormal blood vessel formation. (**B**). CAR-T cell therapy: CAR-T cells target MM antigens (e.g., BCMA, CD38, GPRC5D), releasing granzymes and perforins to induce MM cell death. (**C**). Bispecific antibody therapy: Bispecific antibodies link T cells to MM cells by binding to CD3 on T cells and MM antigens (e.g., BCMA, CD38, FCRH5, GPRC5D), activating T cells to kill MM cells. Created in BioRender. Zhang, Z. (2025) https://BioRender.com/p98h834 (accessed on 27 January 2025).

**Figure 6 cancers-17-00653-f006:**
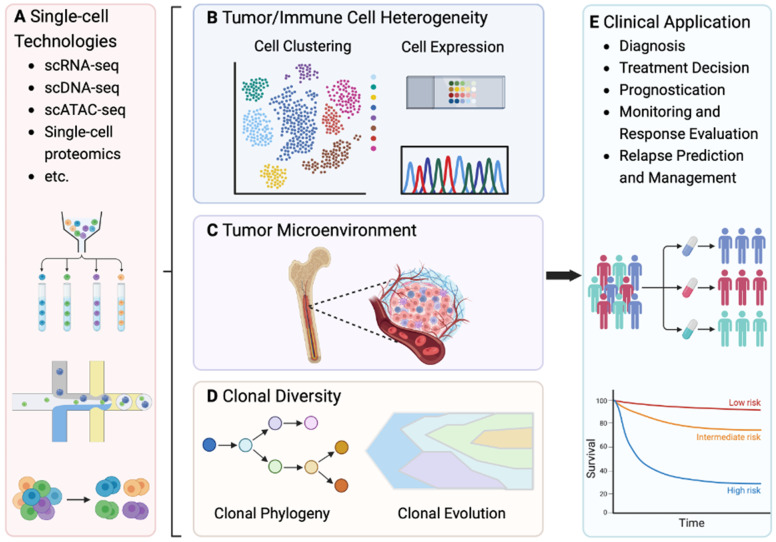
Single-cell technologies and their clinical application. (**A**) Single-cell technologies like scDNA-seq, scRNA-seq, and emerging methods such as scATAC-seq and single-cell proteomics contribute to studying the complexity and potential clinical treatment of MGUS, SMM, and MM. (**B**) The scRNA-seq can be used to identify the dynamics and heterogeneity of both tumor and immune cells by cell-type clustering and expression signature identification. (**C**) Single-cell technologies enable in-depth studies of cellular interactions within the tumor microenvironment. (**D**) Using scDNA-seq can also help to detect treatment responses and drug resistance amid clonal evolution. (**E**) Such early-stage clinical application of single-cell profiling that facilitates its further clinical translation of single-cell technologies includes diagnosis, treatment decision, prognostication, and monitoring of treatment response and residual disease in MM patients. Created in BioRender. Shi, Z. (2025) https://BioRender.com/q34l594 (accessed on 10 February 2025).

**Table 1 cancers-17-00653-t001:** List of relevant single-cell studies used to inform this review. Note that parentheses contain what type of tagging was performed (5′: 5 prime tagging, 3′: 3 prime tagging, NA: not recorded), “+” denotes multiomics on the same cell, and “and” denotes multiomics on different cells potentially from the same sample.

Author	Year	Platform	DOI	REF
Melchor et al.	2014	Fluidigm multiplex qPCR	10.1038/leu.2014.13	[13]
Ledergor et al.	2018	MARS-seq	10.1038/s41591-018-0269-2	[15]
Jang et al.	2019	Fluidigm C1 + MAP-RSeq	10.1038/s41408-018-0160-x	[12]
Zavidij et al.	2020	10x Genomics RNA (3′)	10.1038/s43018-020-0053-3	[38]
Cohen et al.	2021	MARS-seq	10.1038/s41591-021-01232-w	[14]
Liu et al.	2021	10x Genomics RNA (3′ and 5′)	10.1038/s41467-021-22804-x	[30]
de Jong et al.	2021	10x Genomics RNA (3′)	10.1038/s41590-021-00931-3	[23]
Croucher et al.	2021	10x Genomics RNA (NA)	10.1038/s41467-021-26598-w	[19]
Tirier et al.	2021	10x Genomics RNA (3′)	10.1038/s41467-021-26951-z	[34]
Kumar et al.	2022	10x Genomics RNA (NA)	10.1038/s41408-022-00636-2	[27]
He et al.	2022	10x Genomics RNA (5′) + V(D)J	10.1002/ctm2.757	[24]
Liang et al.	2022	10x Genomics RNA (3′)	10.1186/s12943-022-01648-z	[29]
Boiarsky et al.	2022	10x Genomics RNA (3′)	10.1038/s41467-022-33944-z	[16]
Pilcher et al.	2023	10x Genomics RNA (3′) and CITE-seq	10.1038/s41525-022-00340-x	[31]
Chen et al.	2023	10x Genomics RNA (3′)	10.1186/s13578-023-00971-2	[18]
Lannes et al.	2023	10x Genomics DNA (NA)	10.1200/JCO.21.01987	[1]
Yao et al.	2023	10x Genomics RNA (3′)	10.1158/0008-5472.CAN-22-1769	[37]
John et al.	2023	10x Genomics RNA (5′) + V(D)J and 10x Genomics ATAC	10.1038/s41467-023-40584-4	[25]
Dang et al.	2024	10x Genomics RNA (5′) + V(D)J (TCR or BCR)	10.1182/blood-2024-210080 10.1016/j.ccell.2023.05.007	[21] [22]
Poos et al.	2023	10x Genomics RNA (3′) and 10x Genomics ATAC	10.1182/blood.2023019758	[32]
Borsi et al.	2024	Mission Bio Tapestry DNA	10.3390/cells13080657	[39]
Rade et al.	2024	10x Genomics RNA (5′) + V(D)J (TCR) + V(D)J (BCR) + Protein (Ab)	10.1038/s43018-024-00763-8	[33]
Cenzano et al.	2024	10x Genomics RNA (3′)	10.1101/2024.04.24.589777	[17]
Johnson et al.	2024	10x Genomics RNA (NA) + ATAC	10.1038/s41467-024-48327-9	[26]
Koh et al.	2024	Mission Bio Tapestry DNA + Protein	10.21203/rs.3.rs-4672454/v1	[40]
Cui et al.	2024	10x Genomics RNA (3′)	10.1158/1078-0432.CCR-24-0545	[20]
Wang et al.	2024	10x Genomics RNA (3′)	10.1038/s41419-024-07027-4	[36]
Larrayoz et al.	2024	10x Genomics RNA (NA) and V(D)J (TCR)	10.1182/blood-2024-194041	[28]
Avigan et al.	2024	scRNA-seq	10.1182/blood-2024-194277	[41]
Verheye et al.	2024	10x Genomics RNA (3′)	10.1186/s13045-024-01629-3	[35]

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
