# Peer review of "Multiple Myeloma Insights from Single-Cell Analysis: Clonal Evolution, the Microenvironment, Therapy Evasion, and Clinical Implications"

_cancers, 2025, doi:10.3390/cancers17040653_

Round 1
Reviewer 1 Report
Comments and Suggestions for Authors
This review is well written and informative. Single cell technology is important for the planimng of therapy as resistant clones may be attacked earlier than without such technology. However, it does not change trfeatment strategies unless the sensitivity of clonal aberrations is known. Until we know the sensitivity and frequency of single aberrations the singel cell technology is a matter of research. The microenvironmental influence is cited, but not really analyzed. Mesenchymal stromal cells, T cells and M2 macrophages may play an important role in progression of myeloma as well as VEGF and vascularization in order to get a more complete view of the clonal changes. Furthermore it would be of interest whether aberrant clones preexist or develop in response to therapy, some myeloma patients develop AML and vice versa, an indication of the mutation in anearly stem cell.
Author Response
Comment 1: “Single-cell technology is important for the planning of therapy as resistant clones may be attacked earlier than without such technology. However, it does not change treatment strategies unless the sensitivity of clonal aberrations is known. Until we know the sensitivity and frequency of single aberrations, the single-cell technology remains a matter of research.”
Response:
We appreciate the reviewer’s insightful observation regarding the current limitations of single-cell technology in altering treatment strategies. We agree that while single-cell technology enhances our understanding of clonal evolution and resistance mechanisms, its direct clinical impact is still under investigation. To reflect this, we have expanded our discussion in [CLINICAL UTILITY OF SINGLE CELL TECHNOLOGIES IN THE MYELOMA, Page 15, line 578-581], emphasizing the need for further research into the clinical sensitivity and frequency of aberrant clones before single-cell insights can be fully integrated into therapeutic decision-making.
“Third, the sensitivity and prevalence of genetic events identified through single-cell technologies must first be determined before they can be implemented in clinical practice, otherwise the treatment strategies will not be effective.”
Comment 2: ”The microenvironmental influence is cited, but not really analyzed. Mesenchymal stromal cells, T cells, and M2 macrophages may play an important role in the progression of myeloma as well as VEGF and vascularization in order to get a more complete view of the clonal changes.”
Response:
We appreciate this important point regarding the role of the tumor microenvironment in myeloma progression. In response to the reviewer’s comment, we have expanded our discussion in [INTERPLAY BETWEEN SUBCLONES AND THE MICROENVIROMENT, Page 11, line 413-424] to include a more detailed analysis of the influence of mesenchymal stromal cells, T cells, and M2 macrophages in shaping the clonal landscape. Additionally, we have elaborated on the role of VEGF and vascularization in facilitating tumor progression and therapeutic resistance.
“To further elaborate on the role of the tumor microenvironment in MM resistance, we highlight key cell types influencing disease progression. Mesenchymal Stromal Cells (MSCs) provide structural support and contribute to a pro-tumorigenic niche by secreting cytokines and extracellular matrix components that improve myeloma cell survival and therapy resistance [115]. Dysfunctional T-cell immunity, particularly T-cell exhaustion and impaired cytotoxicity, allows for immune evasion and disease progression in MM [116]. M2 macrophages contribute to an immunosuppressive tumor microenvironment by secreting anti-inflammatory cytokines, reducing antitumor immune responses [117]. Vascularization, i.e., enhanced angiogenesis, driven by VEGF and other pro-angiogenic factors, promotes nutrient delivery to myeloma cells, facilitating their proliferation and drug resistance [118]. These factors actively influence the clonal evolution of MM, thus contribute to the intricate and dynamic interactions within its microenvironment.”
Comments 3: “Furthermore, it would be of interest whether aberrant clones preexist or develop in response to therapy. Some myeloma patients develop AML and vice versa, indicating a mutation in an early stem cell.”
Response:
We acknowledge the significance of understanding whether aberrant clones preexist or emerge as a result of treatment pressure. In response, we have included additional discussion in [DISCUSSION, Page 16, line 627-634] addressing it as an open problem in the myeloma field that has not been resolved but we could start to figure it out with single cell.
“It would also be valuable to investigate whether aberrant clones exist prior to treatment or emerge as a result of therapy. The occurrence of acute myeloid leukemia in some myeloma patients, and vice versa, suggests the presence of a mutation in an early stem cell. However, relapsed or refractory patients have increased incidence of high risk mutations and cytogenetics alluding to selection pressure from treatment. This remains an unresolved issue in the field of myeloma, but single-cell technology for patients at multiple timepoints during progression and treatment could provide insights to help address it.”
Reviewer 2 Report
Comments and Suggestions for Authors
This is a well written narrative review aiming to summarize the current impact of single cell technologies in the field of MM.
As the authors stated at the end, a collaborative effort between researchers, clinicians and bioinformaticians is more necessary than ever.
The paper describes single cell technologies and highlights recent studies. The clonal heterogeneity, clonal evolution, basic concepts on treatment resistance, the role of bone marrow microenvironment, and modern immunotherapies are illustrated with the help of well-designed figures. The core of the manuscript is the clinical utility of these technologies and the associated challenges for its translation into clinical practice. In the discussion the authors point out some crucial steps for
achieving this goal.
Now most patients are clinically managed with FISH and the help in some sites of NGS. This represents a step forward.
Some minor comments:
Despite recent advances, MM remain an incurable disease. I think everybody involved in the study of this disease should be aware of this. I suggest including this adjective in the initial definition (line 35).
Table 1: only the “year” could be placed in the Date column, for simplicity and best table fit.
Line 113: The term “precursor diseases” is more appropriated that “early stages” in reference to MGUS and SMM.
Line 134: regarding “treatment resistance” associated to 1q+ and translocations, please add “in some studies”. Anyway, a reference to specific therapy is needed, provided that with the use of modern therapy approaches, this seems to be surpassed. As the authors stated on line 156, quad therapies are now standard, particularly D-RVD in transplant-eligible NDMM.
A comment about the difference between 1q gain and 1q amp could be recommended.
The prognostic impact of single 1q abnormalities by FISH has been largely explored. There are some meta-analysis available. In this context, the importance of other cytogenetic abnormalities (double and triple-hits patients) should be mentioned.
Overall, prospective studies and trials such as NCT04065789 are needed to accurately evaluate the role of these technologies. High quality epidemiological, clinical, and imaging (…) data must be comprehensively recorded and integrated with sc technology dynamic data.
Figure 4: text resolution is poor, consider using abbreviations to improve readability.
No word about artificial intelligence help is commented…
Author Response
Comments 1: “Despite recent advances, MM remain an incurable disease. I think everybody involved in the study of this disease should be aware of this. I suggest including this adjective in the initial definition (line 35).”
Response:
Thank you for pointing it out. We agree with this comment. Therefore, we have emphasized the point that MM is still incurable in [ABSTRACT, Page 1, Line 40-41].
“MM remains an incurable disease.”
Comments 2: “Table 1: only the “year” could be placed in the Date column, for simplicity and best table fit.
Response:
Agree. We have accordingly modified Table 1 to emphasize this point in [Table 1, Page 3, Line 97-98].
Comments 3: “Line 113: The term “precursor diseases” is more appropriated that “early stages” in reference to MGUS and SMM.”
Response:
Thank you for pointing it out. We agree with this comment. Therefore, we changed “early stages” to “precursor diseases” in [SUBCLONALITY AND TUMOR EVOLUTION, Page 4, Line 128].
“This subclonal architecture is evident even in precursor diseases,”
Comments 4: “Line 134: regarding “treatment resistance” associated to 1q+ and translocations, please add “in some studies””
Response:
Agree. We have accordingly added “in some studies” in [SUBCLONALITY AND TUMOR EVOLUTION, Page 5, Line 149].
Comments 5: “Anyway, a reference to specific therapy is needed, provided that with the use of modern therapy approaches, this seems to be surpassed.”
Response:
Thank you for pointing it out. We agree with this comment. Therefore, a reference to specific therapy is added in [SUBCLONALITY AND TUMOR EVOLUTION, Page 5, Line 151].
“Moreover, the presence of multiple, genetically distinct subclones complicates treatment, as different subclones may harbor resistance mechanisms against various therapies, such as Bortezomib.”
Comments 6: “The prognostic impact of single 1q abnormalities by FISH has been largely explored. There are some meta-analysis available. In this context, the importance of other cytogenetic abnormalities (double and triple-hits patients) should be mentioned.”
Response:
Agree. The importance of other cytogenetic abnormalities (double and triple-hits patients) is added in [SUBCLONALITY AND TUMOR EVOLUTION, Page 7, Line 233-239].
While single 1q abnormalities have been widely studied in multiple myeloma, recent analyses suggest that patients harboring double- or triple-hit cytogenetic abnormalities, such as concurrent 1q amplification, del(17p), and t(4;14), exhibit significantly poorer prognoses. These high-risk patients tend to exhibit elevated genomic instability, making their disease more aggressive. As a result, they often respond poorly to standard therapies, thus require alternative treatment approaches and more vigilant clinical monitoring [84].
Comments 7: “Overall, prospective studies and trials such as NCT04065789 are needed to accurately evaluate the role of these technologies. High quality epidemiological, clinical, and imaging (…) data must be comprehensively recorded and integrated with sc technology dynamic data.”
Response:
Thank you for pointing out need for prospective studies and trials to fully assess the clinical impact of single-cell technologies. We have highlighted the importance of well-designed prospective studies, such as NCT04065789, in validating the clinical utility of these technologies in [DISCUSSION, Page 16, Line 643-648]
“Future prospective studies and clinical trials, such as NCT04065789, will play a crucial role in fully uncovering the impact of single-cell technologies in precision oncology. By integrating high-resolution single-cell data with comprehensive epidemiological, clinical, and imaging information, researchers can gain a more in-depth understanding of disease heterogeneity, the evolution of resistance mechanisms, as well as potential therapeutic targets.”
Comments 8: “Figure 4: text resolution is poor, consider using abbreviations to improve readability.
Response:
Agree. The text resolution in Figure 4 has been adjusted for better visual presentation.